# Enlightening the Pathway of Phytoremediation: Ecophysiology and X-ray Fluorescence Visualization of Two Chilean Hardwoods Exposed to Excess Copper

**DOI:** 10.3390/toxics10050237

**Published:** 2022-05-06

**Authors:** Estefanía Milla-Moreno, Robert Dean Guy, Raju Y. Soolanayakanahally

**Affiliations:** 1Department of Forest and Conservation Sciences, Faculty of Forestry, University of British Columbia, Forest Sciences Centre, 2424 Main Mall, Vancouver, BC V6T 1Z4, Canada; rob.guy@ubc.ca; 2Indian Head Research Farm, Agriculture and Agri-Food Canada, Indian Head, SK S0G 2K0, Canada; raju.soolanayakanahally@agr.gc.ca

**Keywords:** synchrotron radiation, photosynthesis, native species, tailings, Chile

## Abstract

In the present climate emergency due to global warming, we are urged to move away from fossil fuels and pursue a speedy conversion to renewable energy systems. Consequently, copper (Cu) will remain in high demand because it is a highly efficient conductor used in clean energy systems to generate power from solar, hydro, thermal and wind energy across the world. Chile is the global leader in copper production, but this position has resulted in Chile having several hundred tailing deposits. We grew two Chilean native hardwood species, quillay (*Quillaja saponaria* Molina) and espino (*Vachellia caven* (Molina) Seigler & Ebinger, under three increasing Cu levels (0, 50, and 100 µM) for 6 months in a greenhouse setting. We measured growth, photosynthetic performance and elemental contents of leaves and roots to further evaluate their potential for phytoremediation. Growth of quillay was unaffected by Cu treatment but growth of espino was enhanced, as was its photosynthetic performance, indicating that espino may have an unusually high requirement for copper. Excess Cu was mostly restricted to the roots of both species, where X-ray fluorescence (XRF) mapping indicated some tendency for Cu to accumulate in tissues outside the periderm. Calcium oxalate crystals were prominently visible in XRF images of both species. Nickel (but not Cu) showed a concurrent distribution pattern with these crystals.

## 1. Introduction

Evidence shows that human influence has warmed the atmosphere, ocean, and land. The last IPCC report indicated that, among other things, changing temperatures have already led to shifts in climates zones towards the poles and an average two days extension of the growing season, significantly affecting vegetation around the globe [1]. Therefore, areas with high biodiversity should be protected.

Central Chile is one of 36 hotspots of biodiversity in the world, and is one of 25 hotspots that together account for only 1.4% of the land surface of the planet but in which 44% of all vascular plant species are confined [2]. Conservation priorities should ensure protection of these highly biodiverse and threatened ecosystems, but climate change may force their displacement. For example, Muñoz-Sáez et al. [3] reported that sclerophyllous forest in Chile is at a higher level of climate risk under global circulation models by 2080, with 65% of high risk in non-protected areas (13,675 km^2^) and 30% in protected areas (238 km^2^).

Central Chile’s flora overlaps with the Chilean mining sector, due to the extensive amount of copper ores present in that region. The rich quality of these ores paved the way for Chile’s historic role as the main copper producer globally. However, because of incomplete extraction, about 450 thousand USD are lost daily in the copper industry as slag resulting from the smelting process [4]. The modest level innovation of the industry (e.g., lack of renewable energies across facilities, recycling of copper from slags, tailings management), and a relaxation of the required mean ore quality required to extract the mineral from 0.87% in 2007 to 0.65% in 2016 [5], has increased the number of tailings in the country.

Copper mines will continue to exist because copper is a crucial element in clean energy systems and its use could balance out the cost from declining ore quality [6], particularly in high solar radiation and copper-rich locations like Chile and Peru, when using renewable energy for copper mining. Copper mining countries (e.g., Chile, Peru, United States) should aim for a fully renewable energy supply by 2030. Zaldívar is the first such fully renewable mine in the world, located in Antofagasta, Chile [6]. To further minimize environmental impact, restoration of mining sites (e.g., tailings) must also be a priority. In this context, the use of plants and their associated microbes for in situ site remediation, a concept coined phytoremediation [7], has been used around the world with great success, particularly when using native species. Broadly defined, phytoremediation may involve one or more of phytoextraction, phytodegradation, rhizofiltration, phytostabilization, phytovolatilization and even the use of plants to filter air pollutants [8,9].

Plants deal with high concentrations of heavy metals, via mechanisms of tolerance, avoidance, or both [10,11]. The best species choice for any given phytoremediation application will depend on the strategy of the plant. For example, a study on metallophytes growing on a copper mine tailing in northern Chile [12] found that *Schinus polygamus* (Cav.) Cabrera and *Atriplex deserticola* Phil., both Cu-hyperaccumulators, accumulated more than 0.1% Cu in their shoots, showing good potential for phytoextraction; whereas *Casuarina equisetifolia* L. accumulated even more Cu, 2.9 g kg^−1^, but in the roots, making it an appropriate candidate for phytostabilization. In cases where phytostabilization rather than phytoextraction is the goal, the preferred plants should avoid metal mobilization to aerial parts where they may be subject to animal browse and/or result in the unwanted dispersal of heavy metals in the leaf litter.

There have been several studies that evaluated the copper tolerance of native Chilean species growing in (or in proximity to) sites with a high level of copper and other metals [12,13,14,15,16,17]. These studies show a wide range of options for phytoremediation, but the hope is to restore those sites to a level of complexity that includes a great variety of microorganisms, flora and fauna, aiming to reflect complex interactions between diversity, structure and ecological processes, as advised by Ruiz-Jaen and Aide [18]. Since the aim of ecological restoration is to assist the recovery of an ecosystem, species known to exist in those sites prior to the disturbance are likely to have better possibilities for survival, growth and reproduction, over species that were not part of that landscape. In this context, and considering that invasive species reduce biodiversity and alter ecosystem functions, measures to promote native plants have been advised [19], particularly if they are identified as vulnerable or endangered [15]. Moreover, native plants that have added value (e.g., are medicinal, melliferous, good for forage, etc.) should be a priority [15], due to their increased benefit to local communities.

Research is needed to aid decision-makers in the mining sector in selecting the best candidates for phytoremediation. A major challenge to using the many native and endemic plant species that are potentially available is the lack of information regarding their propagation, basic physiological requirements, tolerance levels, metal uptake and sequestration patterns, etc. Assessment of metal distribution in tissues is crucial to understanding environmental impacts, tolerance mechanisms, and interactions between multi-metal and other stressors in plants. One relatively fast and non-destructive way to survey multi-element distributions in plant tissues is via X-ray fluorescence (XRF) as described by Vijayan [20]. This technique has been proven versatile for visualizing copper bioaccumulation in algae [21], studying copper tolerance of willow in flooded soils [22], and monitoring zinc distribution in drought-stressed leaves of wheat (*Triticum aestivum* L.) under drought stress [23].

We set out to explore the ecophysiology and metal sequestration patterns of two species native to sclerophyllous forests in Chile, both with great potential for tailings restoration; namely, quillay (*Quillaja saponaria* Molina) and espino (*Vachellia caven* (Molina) Seigler and Ebinger, formerly assigned to *Acacia*). Espino (‘Kawen’: paddle in Mapüdungun), a member of the *Fabaceae*, is a nitrogen-fixing, semi-deciduous shrub or small tree (up to 6 m) with rigged thick bark, spinescent stipules, and bipinnately compound leaves that are ultimately dissected into tiny leaflets about 1–2 mm long and 0.5 mm wide [24]. Espino has been intensively used for charcoal production, due to its high caloric content, and its protein-rich seeds and pods are valued as forage [24]. In contrast, quillay (‘Küllay’: a certain tree in Mapüdungun), a member of the *Quillaceae*, is an evergreen tree (to 20 m) with thick rounded leaves that are 3–4 cm long and 1.5–3 cm wide [25]. Quillay provides good cover for foraging animals and is a rich source of saponins with a variety of industrial applications [25]. We compared seedling growth of these two species in a greenhouse experiment under two levels of Cu stress and, after 6 months, we measured photosynthesis, chlorophyll fluorescence (as a stress indicator) and elemental concentrations, and performed XRF mapping of metal distributions. In a previous field study [16], we found that several native species, espino and quillay included, tolerated the stress of various metals in the tailings for six years with no clear advantage from amendments (i.e., mycorrhiza and compost). Moreover, copper was prominent in quillay roots but not leaves, whereas in espino there was less copper uptake overall and it was more evenly distributed. We hypothesized that root versus shoot copper distribution patterns would differ in espino and quillay under controlled environment conditions, possibly leading to differential effects on photosynthesis in these two species.

## 2. Materials and Methods

### 2.1. Plant Material and Treatments

Seeds of espino and quillay were obtained commercially (Vivero Antumapu, Chile) and planted in 1-gallon pots containing a 70% peat moss and 30% perlite mixture in a completely randomized design at the University of British Columbia greenhouse between December 2018 and June 2019. Espino seeds needed chemical scarification with sulfuric acid to break dormancy [25]. Quillay seeds were stratified by soaking in cold water for 72 h [25]. A fertigation system delivered 100 mL of liquid media (Appendix A) to each pot every other day for the first 3 months, and then 150–300 mL each day for the remaining three months, a sufficient quantity to ensure that nutrient solution drained freely from the pots on every watering. Dripper outlets were repositioned bi-weekly. After 2 months to allow plant establishment, additional CuSO_4_ was added to the fertigation solution to achieve copper treatment concentrations of 0, 50 and 100 µM above the normal Cu concentration of 0.0033 µM Cu. There were five pots per species per treatment. Natural light provided a photosynthetic photon flux density (PPFD) of 470 μmol m^−2^ s^−1^ averaged over the entire 6 months. Supplemental lighting to ensure a minimum PPFD of 240 μmol m^−2^ s^−1^ and a 12 h photoperiod was applied. Temperatures averaged 26 °C during the day and 19 °C at night.

### 2.2. Physiological Measurements

A LI-COR 6400 XT portable infrared system (LI-COR, Biosciences, Lincoln, NE, USA) was used to measure gas exchange traits on the youngest mature sun-exposed leaves (three times for each plant) over a 5 week period beginning 13 weeks after treatments began, including maximum photosynthetic rate (Amax; μmol CO_2_ m^−2^ s^−1^), stomatal conductance (*g*s; mol H_2_O m^−2^ s^−1^) and instantaneous WUE as determined by Amax over transpiration (WUEi; μmol CO_2_ mmol H_2_O^−1^). Leaves of quillay covered the full window of the cuvette, but the feathery leaves of espino only covered about 22% of the frame (calculated with a leaf area meter and three measurements of an espino leaf on a 6 cm^2^ frame to match the LI-COR). We applied a factor of 4.587 to account for this difference. After gas exchange measurements were completed, the same leaves were then dark-adapted for 20 min and the maximum quantum efficiency of PSII photochemistry (Fv/Fm) was obtained after a saturating flash (>7000 μmol m^−2^ s^−1^) as per Genty et al. [26], from 10:00 to 14:00 hrs with a FluorPen FP 100 (Photon Systems Instruments, Drasov, CZ).

Plants were harvested 18 weeks after treatment initiation, at which time the chlorophyll content index (CCI) was estimated on the youngest mature leaf using a CCM-200 Plus (Opti-Sciences Inc, Hudson, NH, USA). Two types of samples were taken from each plant after washing the potting soil mix from the roots and making basic growth measurements (shoot length, root length and fresh weight). Leafy twigs were pressed and air-dried, and root segments were placed into open vials and dried over silica gel in a desiccator. After further drying at 70 °C, followed by microwave digestion, root and leaf Al (mg kg^−1^), Ca (mg kg^−1^), carbon (%), Cu (mg kg^−1^), Fe (mg kg^−1^), K (mg kg^−1^), Mn (mg kg^−1^), Mo (mg kg^−1^), Na (mg kg^−1^), nitrogen (%), P (mg kg^−1^) and Zn (mg kg^−1^) concentrations were determined by ICP-MS at the Analytical Chemistry Research Laboratory of the British Columbia Ministry of Environment. There was limited material for these analyses for espino; hence, for this species, composite samples consisting of an equal quantity of tissue from all five plants per treatment were constructed. After harvesting, about 300 g of fresh soil mix from each pot was also shipped to the same laboratory for immediate analysis of pH, soil mix organic matter (by loss on ignition), and concentrations of oxalic and citric acid (via water extraction, LC Analysis). Soil mix Ca (mg kg^−1^), Cu (mg kg^−1^), Fe (mg kg^−1^), K (mg kg^−1^), Mn (mg kg^−1^), Mo (mg kg^−1^) and Zn (mg kg^−1^) were determined by ICP-MS as above. The Cu bioconcentration and bioaccumulation factors were calculated in two ways: first as the ratios of the mean Cu concentrations in the roots (*Bcf*_soil mix_) and leaves (*Baf*_soil mix_) to that of the soil mix, respectively, and second as the ratios of the mean Cu concentrations in the roots (*Bcf*_solution_) and leaves (*Baf*_solution_) in mg kg^−1^ to that of the fertigation solution in mg L^−1^ [27], respectively, while the translocation factor (*Tf*) was calculated as the ratio of Cu concentration in the leaves relative to the roots [28].

### 2.3. Element Visualization

X-ray fluorescence microimaging of single leaf and root subsamples from each species grown with (100 μm) and without (0 μM) added Cu was performed at the Canadian Light Source Synchrotron facility (Saskatoon, SK). Dried leaves were placed directly on Kapton tape for imaging. Roots were placed in centrifuge tubes and immersed in millipore water overnight for rehydration. Samples were then taken out of the water and positioned in molds containing cryogel for freezing with liquid nitrogen. After this, radial cross-sections of 80 μm thickness were obtained with a microtome (Leica CM1950, Leica Microsystems), re-dried, then placed on Kapton tape. Images were collected on the BioXAS-Imaging beamline [29] configured as follows: incident energy of 15 keV, micro-mode, stage set at 90 degrees to the incident beam, detector at 45 degrees to the incident beam, beam focus of 5 µm for leaves and 12 µm for roots, with a dwell time of ~100 min. The data processing and mapping were done using PyMCA [30].

### 2.4. Statistics

Soil mix, growth and physiological statistics were analyzed using 2-way ANOVAs in R [31]. One-way ANOVA was used to explore situations where there was a significant interaction in the absence of a major treatment effect. We employed the Bonferroni correction when we analyzed 2-way ANOVA in growth and photosynthesis using R. For leaf and root elemental concentrations and heavy metals, and because tissues were pooled for espino, t-tests were conducted to evaluate differences between species and 1-way ANOVA was used to examine treatment effects in quillay.

## 3. Results

### 3.1. Soil Mix

We analyzed the soil mix in three pots without plants. Copper content averaged 15, 203 and 354 mg kg^−1^ in the control (hereafter referred to as 0 µM) and 50 µM and 100 µM Cu treatments, respectively. The soil mix was slightly acidic (5.7 to 6.0), with about 50% organic matter content and average oxalic and citric acid concentrations of 267 and 19 mg kg^−1^, respectively. Calcium, K and P accounted for 5–9%, 0.7–1% and 0.2–0.3% of the soil mix dry mass, while concentrations of Fe, Mn, Mo and Zn ranged from 3–5 g kg^−1^, 0.13–0.15 g kg^−1^, 7–10 mg kg^−1^ and 30–34 mg kg^−1^, respectively.

Soil mix pH, Cu, Fe, K, Mn, Mo and organic acid contents were modified by the presence of plants, depending on species and/or treatment (Figure 1, Appendix A). Soil mix with plants rooted in it had somewhat less Cu, Mn and Mo, approximately doubled citric acid and slightly higher K and pH. Soil mix oxalic acid content was increased about 2.4-fold by the presence of quillay but not espino. We found a significant species effect for Cu, Fe, Mo, pH, and organic acids, and a significant treatment effect for Cu, as expected, but also for Mn, Mo, and pH (Appendix A). There was a species by treatment interaction for Cu and pH. Soil mix Mo was reduced in the presence of additional Cu. Espino reduced the soil mix Cu and Mo contents more than quillay, but caused a greater rise in pH in the copper treatments. Pots with espino also tended to have slightly higher Fe content. With both espino and quillay, soil mix in the 50 μM Cu treatment had significantly lower Mn than either of the other treatments.

### 3.2. Growth

Biomass (FW) of quillay exceeded espino by approximately 4-fold when averaged across treatments (Figure 2, Appendix A), but shoot lengths were more comparable. Shoot lengths of espino and quillay, increased significantly as Cu was increased (Appendix A). Similarly, the Cu treatments resulted in a slight increase in FW of espino (*p* < 0.009; Appendix A) and the opposite in quillay (not significant). There were no significant treatment or species effects on root length.

### 3.3. Photosynthesis

Chlorophyll content index was much higher, by roughly nine times, in quillay than in espino (Figure 2d, Appendix A). Two-way ANOVA (Appendix A) indicated no significant effect of treatment, but there was significant interaction with species and, when considered alone, CCI in espino increased significantly with Cu treatment (*p* < 0.001, Appendix A), whereas it trended in the opposite direction in quillay (*p* = 0.080). Likewise, although Amax was unchanged with Cu treatment in quillay (Figure 3a, Appendix A), in espino at 100 μM it was nearly tripled over the control level (*p* < 0.001). Overall, and in contrast to CCI, both Amax and *g*s (Figure 3a,b) were considerably higher in espino than quillay. There were no significant effects of treatment on *g*s or WUEi in either species, but quillay had a significantly higher WUEi than espino (Appendix A, Figure 3c). Two-way ANOVA indicated no effect of species or treatment on QY, but again there was a significant interaction and, when considered alone, one-way ANOVA (Appendix A) showed a significant increase in QY for espino (*p* < 0.003) but not quillay.

### 3.4. Elemental Analysis

Root and leaf elemental concentrations are presented in Table 1. The pooling of tissue samples for espino precluded a 2-way ANOVA, but Cu was clearly increased in both leaves and, more so, roots in rough proportion to the Cu supplied. In quillay, foliar Cu rose from 2.2 mg kg^−1^ in controls to ~10 mg kg^−1^ in Cu-treated plants (*p* = 0.007; Appendix A), and from 12 mg kg^−1^ to 67 and 97 mg kg^−1^ in roots at 50 and 100 μM Cu, respectively (*p* < 0.001). No other elements in leaves were affected by the provision of additional Cu, but Ca (*p* < 0.004) and P (*p* < 0.003) were lower in roots at 50 μM than at 0 or 100 μM Cu.

Although foliar Cu concentrations and bioaccumulation factors tended to be somewhat lower in quillay than in espino, they were not significantly different. Bioaccumulation factors relative to the soil mix (*Baf*_soil mix_) fell considerably when the Cu was increased above the control (Appendix A). Given the very low Cu concentration in the fertigation solution at 0 μM Cu, calculation of the bioaccumulation and bioconcentration factors relative to the fertigation solution (*Baf*_solution_ and *Bcf*_solution_) is not relevant for this treatment. In espino at 50 and 100 μM Cu, the *Baf*_soil mix_ was 0.080 and 0.058 whereas the *Baf*_solution_ was 2.74 and 2.05, respectively. The respective bioconcentration factors were 0.860 and 0.752 (*Bcf*_soil mix_) and 29.3 and 26.8 (*Bcf*_solution_). In quillay at 50 and 100 μM Cu, respectively, the *Baf*_soil mix_ was 0.051 and 0.035 and the *Baf*_solution_ was 3.18 and 1.48. Quillay, however, accumulated significantly less Cu in roots than did espino at both 50 and 100 μM Cu (*p* < 0.02; Appendix A) and consequently also had lower bioconcentration factors in these treatments, with the *Bcf*_soil mix_ being 0.336 and 0.360 at 50 and 100 μM Cu, while the *Bcf*_solution_ was 21.1 and 15.2, respectively. Mean translocation factors were similar in quillay and espino, dropping from 0.177–0.254 at 0 μM Cu, to 0.076–0.097 at 100 μM Cu. At all treatment levels, quillay had only about half as much foliar N as espino (*p* < 0.001), and about 30% less N in the roots. Quillay also always had lower concentrations of Fe in both leaves and roots (*p* < 0.001). Molybdenum too, was lower in quillay than espino in leaves (*p* < 0.001) and in roots (*p* < 0.02), except in roots at 100 μM Cu (*p* = 0.496). In contrast, quillay accumulated more Na in roots than espino (*p* < 0.02), and more K in leaves (*p* < 0.004).

### 3.5. X-ray Fluorescence Microimaging

X-ray fluorescence analysis was restricted to just eight specimens, including single leaf and root samples from each of four plants (one of each species at 0 and 100 μM Cu). The emission spectra from these samples are compared in Figure 4, where in all four panels the total counts are normalized to the control (0 μM) to account for signal strength variation due to sample size and/or total elemental content. Roots have a quite different general profile relative to leaves, suggesting that mobile elements were in large part leached from the roots when they were hydrated for purposes of sample preparation. Hence, peaks for P, S, Cl, K and Mn (at 2.014, 2.308, 2.622, 3.314 and 5.899 keV, respectively) are lower than expected based on our elemental analysis of tissues. The same is also very likely true for bromine, which accounts for the large peak at 11.924 keV in leaf tissues, but which is much smaller in roots. As expected, and noting that counts are plotted on a log scale, the Cu peak is much higher in leaves of both species when they were exposed to elevated copper in the fertigation solution (Figure 4a,b). The copper peak is also more pronounced in the root section of the Cu-treated quillay (Figure 4d), but not so much in espino (Figure 4c). In espino, the leaf sample from the Cu-treated plant shows markedly reduced fluorescence signals from silicon and nickel (arrows in Figure 4a). The root sample of Cu-treated espino shows a similar effect on peaks for vanadium and chromium, and there may be a similar effect of excess Cu on iron and zinc in the root of quillay.

Images for select elements associated with these spectra, in plants receiving 100 μM Cu, are presented in Figure 5 and Figure 6 (leaves), and Figure 7 and Figure 8 (roots). Calcium is shown in all four figures and we include K in the leaf images but not the roots, where instead we show the distribution of Mn. Copper was not restricted to a particular tissue in the leaves of either species, other than a bright spot in the espino sample that might represent contamination. Calcium was also found in concentrated form as calcium oxalate crystals, particularly near the veins of both species. In espino, Ca also accumulated in the pulvinus at the base of each of the two secondary leaflets partially visible in the image.

In roots of espino (Figure 7), Cu was distributed throughout but was particularly concentrated in tissues external to the phloem. Manganese had a similar distribution but was more evenly distributed across the inner (living phloem) and outer bark (periderm and remnant cortical and epidermal tissues), as was Ca. In quillay (Figure 8), the distribution of Cu clearly shows the structure of the xylem, with some increased presence near the cambium, but it is particularly abundant in the outer bark, presumably outside the periderm. Manganese had a similar pattern of localization. In contrast, Ca was most prominent in the phloem tissue near the cambium.

Nickel was easily visualized in the leaf of espino and the root of quillay (Figure 9) and in both cases its distribution was similar to Ca. Co-occurrence of Ni with the pattern of cells containing calcium oxalate crystals was evident in the espino leaf. There may be a similar association between Ni and Ca in the roots of quillay, but the image resolution is insufficient to clearly indicate the presence of calcium oxalate crystals.

## 4. Discussion

Many studies have shown that excess Cu negatively impacts the growth of most plants, often at concentrations of less than 50 μM Cu [32]. For example, exposure of common bean (*Phaseolus vulgaris* L.) to 50 μM Cu resulted in chlorosis, necrosis and a severe reduction in dry weight after a mere 3 days [33]. In contrast, the growth of quillay was not affected after 4 months growth in our 50 and 100 μM Cu treatments, whereas shoot growth in espino was even enhanced. These observations are consistent with the ecologies of these two species and their excellent potential for use in phytoremediation. Total copper in Cu-affected native soil (in Central Chile) ranges from 30 to 200 mg kg^−1^ [34] and 70–155 mg kg^−1^ in agricultural soils of north-central Chile not affected by mining [35]. In Canada, the maximum allowable total Cu concentration in agricultural soils is 63 mg kg^−1^ [36]. These ranges were bracketed by the conditions we imposed, which resulted in final average Cu concentrations in the soil mix of 15, 203 and 354 mg kg^−1^ in the control (0 μM), 50 and 100 μM Cu treatments, respectively. Total Cu in tailings, however, can be much higher; e.g., from 360 to 4000 mg kg^−1^ in tailings of Central Chile [37]. Further, our soil mix was high in organic matter (about 50%) relative to most tailings, which should have the effect of ameliorating metal stress. Clearly, much of the Cu provided in the fertigation solution was bound by the soil mix and its Cu concentration must have increased over the course of the experiment.

Organic matter and pH are known to regulate copper bioavailability and mobilization to aerial parts [38,39]. Low molecular weight organic acids also have a profound effect on metal availability and toxicity [40]. Analysis of our soil mix revealed no effect of the Cu treatments on citric or oxalic acid levels, but oxalic acid was consistently higher in pots occupied by quillay, while citric acid tended to be higher in pots with espino. Although not detected here, plant roots often show increased exudation of organic acids when exposed to heavy metals. For example, Montiel-Rozas et al. [41] found that *Malva sylvestris* exuded oxalic acid in response to a combined Cu, Cd and Zn stress, and Qin et al. [42] reported a similar response to Cu stress in *Populus tremula* L. The binding of metals by organic acids exuded from roots may help in mobilization and increase metal uptake [43] or, contrastingly, limit uptake [44].

Although several species have been defined as Cu-accumulators such as *Ammania baccifera* L. [45], *Oenothera affinis* Cambess. [46], *Mimulus luteus* L. var. *variegatus* (Lodd.) Hook., *Cenchrus echinatus* L., *Erigeron* L., *Mullinum spinosum* (Cav.) Pers., *Nolana divaricate* (Lindl.) I.M. Johnst., and *Dactylium* sp. [47], and *Adesmia atacamensis* Phil. [48], studies on quillay and espino are scarce [16,49]. As in our previous field study [16], we found that Cu accumulation in quillay was more pronounced in roots than in leaves. The translocation factors were similar between the two studies, averaging 0.07 in the field and 0.10 in the present study at 100 μM Cu, underscoring the potential of this species for phytostabilization. Although we reached Cu concentrations in our soil mix that greatly exceeded those of the tailings, Cu concentrations in roots and leaves of quillay were comparable between the two studies. In the greenhouse, however, quillay did not accumulate Cu from the fertigation solution any faster than it was bound by the peat-based soil mix and the *Bcf*_soil mix_ remained below 1.0 regardless of the treatment (Appendix A). Consequently, from this perspective, the bioconcentration factor was much higher on the tailings than in the greenhouse, by as much as 36-fold. Likewise, the bioaccumulation factor was also considerably higher on the tailings as compared to the *Baf*_soil mix_ that we found here. It is likely that the measurement of total Cu in the tailings and the soil mix (as opposed to bioavailable Cu) is behind these differences. We don’t know the bioavailability of the Cu in the tailings but the Cu concentration of the fertigation solution, though mitigated by the soil mix, provides a benchmark for the greenhouse study. From this other perspective, the field and greenhouse results are more comparable, with the *Bcf*_solution_ and *Baf*_solution_ averaging 15.2 and 1.48, respectively, for quillay at 100 μM Cu. Regardless, the bioconcentration and bioaccumulation factors from the field are more relevant for applied purposes, but unlike tissue concentrations and elemental distributions they provide little physiological information.

In contrast to quillay, the distribution of Cu between the roots and leaves of espino was quite different here as compared to the field. In the greenhouse espino behaved like quillay and restricted Cu translocation to the leaves (*Tf* = 0.08 at 100 μM Cu), but in the field Cu was more evenly distributed between roots and leaves (mean *Tf* = 0.94). This inconsistency prevents us from recommending espino for purposes of phytostabilization without further study. We noted a few other differences in elemental composition between espino and quillay, independent of the treatments imposed. For example, Mo was higher in espino than quillay, which might be related to a greater need for this element in support of symbiotic nitrogen fixation via *Rhizobium* bacteria. Molybdenum acts as a cofactor for nitrogenase during the conversion of N_2_ into ammonium [50]. In contrast, across treatments and tissues the Na content of quillay was much higher than in espino (Table 1).

There were large differences in Amax and *g*s between espino and quillay. These differences may in part relate to Amax and *g*s being over-estimated in espino because of difficulty in determining leaf area within the cuvette of the gas exchange system. The leaves of espino are finely dissected and sensitive to touch, tending to fold when they are disturbed. In contrast, the CCI was considerably higher in quillay than in espino. This difference may be related to the evergreen nature of quillay leaves, but also likely again reflects the difference in leaf anatomy causing espino leaves to not fully cover the device aperture. Espino seemed to be lacking copper for optimal photosynthetic performance, with Amax peaking in the 100 μM Cu treatment, whereas in quillay Amax remained stable across treatments. Although foliar nitrogen, Amax and *g*s were higher in espino than in quillay across all treatments, quillay accumulated a greater fresh mass.

Unlike Amax and *g*s, calculations for WUEi and QY are independent of leaf area. The WUEi of espino seemed to increase with copper while in quillay it was more stable and significantly higher. Similarly, in espino, QY, CCI, shoot length and biomass (FW) were all significantly higher when copper was added. We conclude that espino has a requirement for copper that is much higher than typical and suggest that it may be an obligate metallophyte [51]. Nonetheless, regardless of treatment, the foliar Cu in both species remained below the typical toxicity threshold for leaves of average plants (i.e., 20 mg kg^−1^ [39]) and the allowable threshold for animal feed (i.e., 40 mg kg^−1^ [52]).

X-ray elemental analysis and heavy metal mapping can shed light on the ecophysiology of metal accumulators and their tolerance and/or avoidance strategies [53]. This technique allowed us to analyze the distribution of many elements, several of which are not shown here. As noted in the results, however, the rehydration protocol necessary for making root sections appears to have leached some mobile constituents, leaving fixed or bound fractions behind. This effect will depend on the element. For example, K and Ca are normally abundant in plant tissues, given that K^+^ is the major free cation in plant cells [54], whereas Ca is structurally important in cell walls and for membrane stability through its interactions with pectin and phospholipids, respectively [55]. Hence, rehydration is expected to affect the distribution of K more than Ca. We used K and Ca to highlight leaf structure in Figure 5 and Figure 6, and Ca and Mn for roots in Figure 7 and Figure 8. Another element with a strong XRF signal in leaves, but not roots, was bromine (Figure 4). Bromine was not supplied in the fertigation solution but is commonly found in peaty soils and readily taken up by plants [56]. Emission spectra suggested that excess Cu may have affected levels of Si and Ni in the espino leaf, and other elements in the roots (V and Cr in espino, and Fe and Zn in quillay), but further work is needed to follow-up on these observations. Copper was shown to competitively inhibit Ni uptake and reduce transport to the shoot in *Glycine max* (L.) Merr. [57].

Copper was widely distributed across both leaf and root tissues, with some tendency to reach higher levels (with Mn) in what we presume to be dead cortical tissue and/or the periderm of roots (Figure 7 and Figure 8). Copper is an essential plant micronutrient, acting as a catalyst in the photosynthetic and respiratory electron transport chains, and as a cofactor for numerous other enzymes. It is substantially involved in the control of the cellular redox state (e.g., via the activity of Cu/Zn superoxide dismutase) and in the formation of lignin in cell walls [58]. This latter role may account for the thin band of Cu in the vicinity of the cambium in the root of quillay (Figure 8). The root sections of both species also show the presence of Cu in the xylem. It is most clearly defined in quillay where it appears to outline the pattern of the cell walls, except in the wood rays. Cell walls are major sites of accumulation for heavy metals where they interact with -COOH, -OH and -SH groups, such as the carboxyl-groups of homogalacturonans [59].

Mechanistically, the differences between the species in Cu concentration and distribution may relate to differences in transporter proteins that limit uptake and/or affect the redistribution of Cu to shoots or, for example, root cortical tissue. There could be a role for the root cortex and/or epidermis in preventing the entry of excess copper into the stele. Preferential localization of heavy metals in root epidermal and cortical tissues has been noted in other species [60,61,62]. These tissues die soon after the initiation of secondary growth and the development of the periderm, thus ridding the plant of the sequestered metals. Further studies are needed at different stages of root growth to investigate this as a possible stress avoidance strategy in espino and quillay. Plants may also deal with heavy metals by sequestering them within the vacuole or cell wall, often with the help of chelators such as phytochelatins, metallothioneins, and organic and amino acids [10,58,59,63,64]. Further work is required to elucidate if and where such mechanisms may operate in quillay and espino.

Crystals of calcium oxalate, which is minimally soluble in water (6.1 mg L^−1^ at 20 °C [65]), were strikingly obvious by XRF imaging in both species. Their presence in quillay has been previously reported [66], but, to the best of our knowledge, not in espino. Nevertheless, they are present in many plants and can take various forms [67]. In leaves of quillay, spherical aggregates of crystals (druses) occur throughout the spongy mesophyll but more frequently beneath veins [66], consistent with the pattern seen in Figure 6. Although druses also occur in the cortical parenchyma and pith of quillay stems, elongated column-shaped crystals (styloids) are abundantly interspersed with fibers in the phloem [66] and are likely what we observed in the root section shown in Figure 8.

Calcium oxalate crystals are hidden players in Ca cycling, and a possible heavy metal detoxification strategy [68]. Nickel was not deliberately supplied to the plants but may have been naturally present in the potting mixture or as a trace contaminant of the fertigation solution [38]. In this context, and although we found no clear concurrence with Cu, the distribution of Ni in the espino leaf and quillay root suggested an association with these crystals. Nickel was seen to accumulate near abundant calcium oxalate crystals in the seed coats of *Grevillea rubiginosa* Brongniart & Gris when germinating in the presence of various Ni salts [69].

In conclusion, both quillay and espino demonstrated high tolerance to the provision of excess Cu. Indeed, espino may require considerably more copper than normal to support optimal growth, as also evidenced by its improved photosynthetic performance with increased Cu in the media. Excess Cu was mostly restricted to the roots of both species and showed some tendency to accumulate in tissue layers outside the periderm. Foliar Cu concentrations remained below toxicity levels in both species, in contrast to our previous study of older plants growing on tailings in the field [16], where this threshold was exceeded in espino. These observations suggest that differences in age and/or other soil conditions may affect experimental outcomes. The possible role of calcium oxalate crystals in sequestering heavy metals other than Cu is an interesting avenue for further investigation.

## Figures and Tables

**Figure 1 toxics-10-00237-f001:**
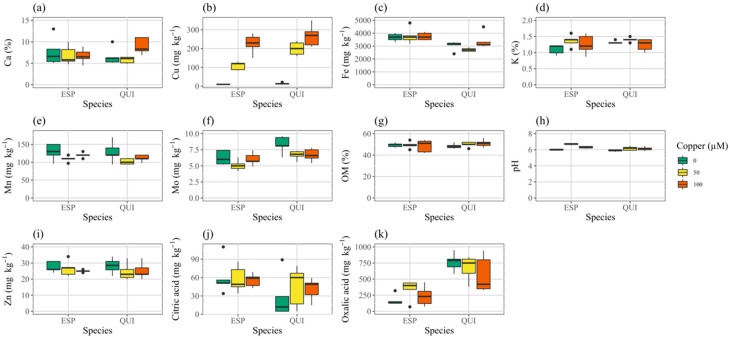
Characteristics of soil mix from pots containing espino or quillay after 6 months growth; (**a**) calcium, (**b**) copper, (**c**) iron, (**d**) potassium, (**e**) manganese, (**f**) molybdenum, (**g**) organic matter, (**h**) pH, (**i**) zinc, (**j**) citric acid, (**k**) oxalic acid. Whiskers in boxplots show the data range excluding outliers.

**Figure 2 toxics-10-00237-f002:**
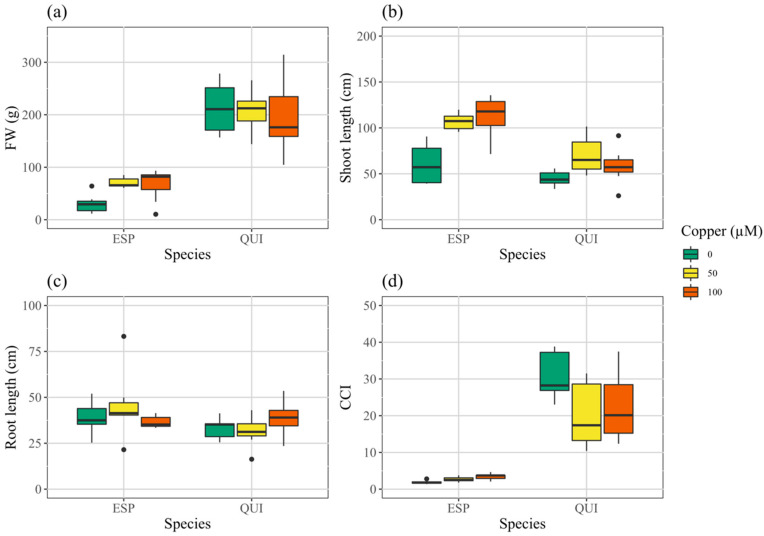
Growth parameters of espino and quillay at time of harvest; (**a**) biomass, (**b**) shoot length, (**c**) root length, and (**d**) CCI.

**Figure 3 toxics-10-00237-f003:**
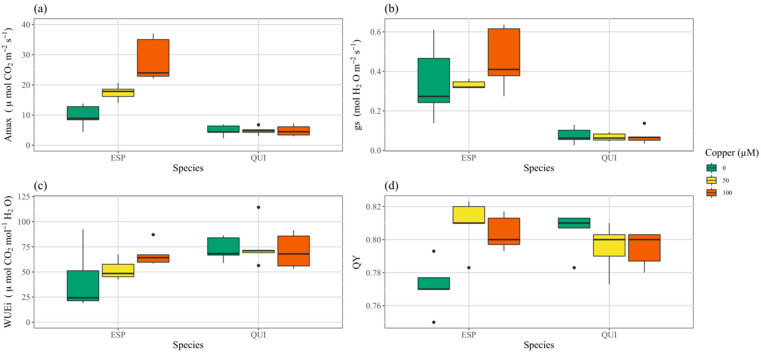
Effects of Cu treatments on photosynthetic traits in espino and quillay; (**a**) maximum photosynthetic rate, (**b**) instantaneous water use efficiency, (**c**) stomatal conductance, (**d**) quantum yield.

**Figure 4 toxics-10-00237-f004:**
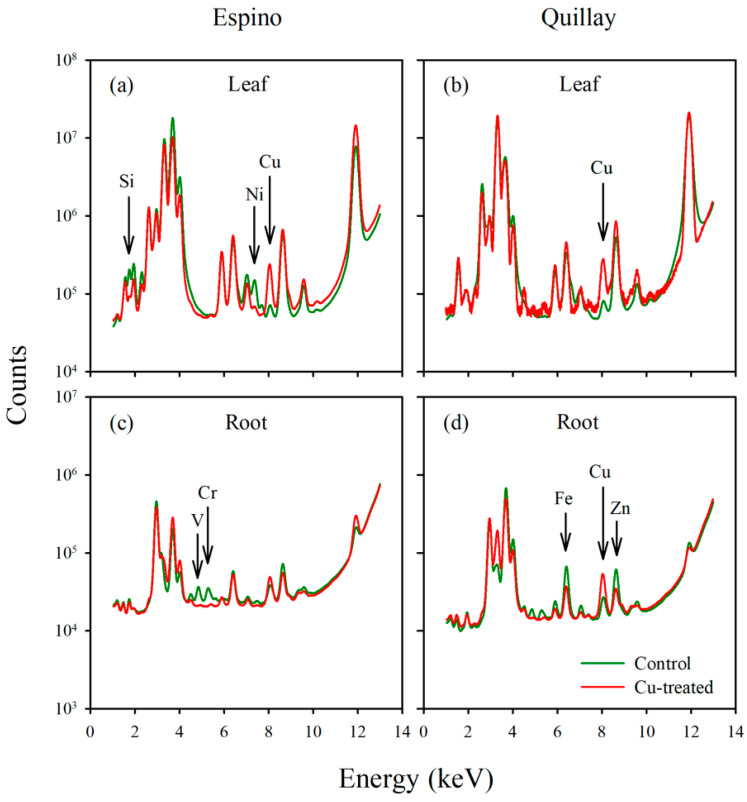
Multi-element spectra in plant tissues; (**a**) espino leaf, (**b**) quillay leaf, (**c**) espino root, (**d**) quillay root. In every panel, the spectrum for Cu-treated tissue (100 μM treatment) is normalized relative to the spectrum for the control (0 μM treatment) by adjusting the total counts upwards or downwards, as appropriate to the sample. This normalization accounts for the lower signal-to-noise ratio for the Cu-treated quillay in panel b.

**Figure 5 toxics-10-00237-f005:**
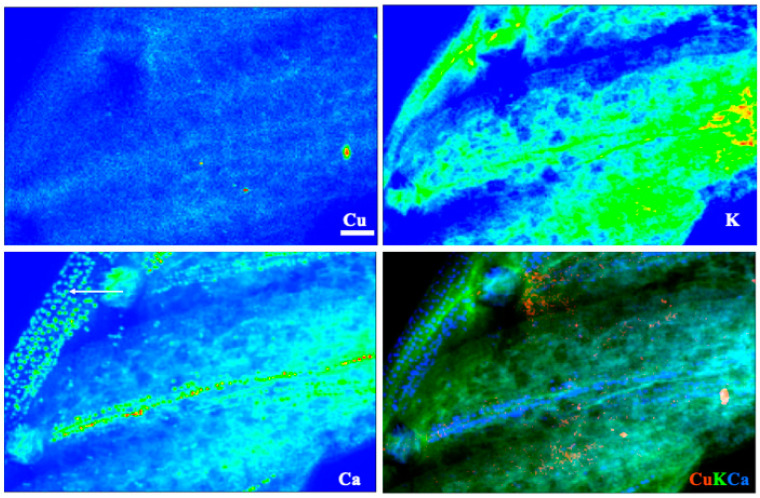
X-ray fluorescence images of leaflets of espino at 100 μm Cu. Scale bar of 100 μm; arrow in the lower left panel showcases the presence of calcium oxalate crystals in the secondary rachis.

**Figure 6 toxics-10-00237-f006:**
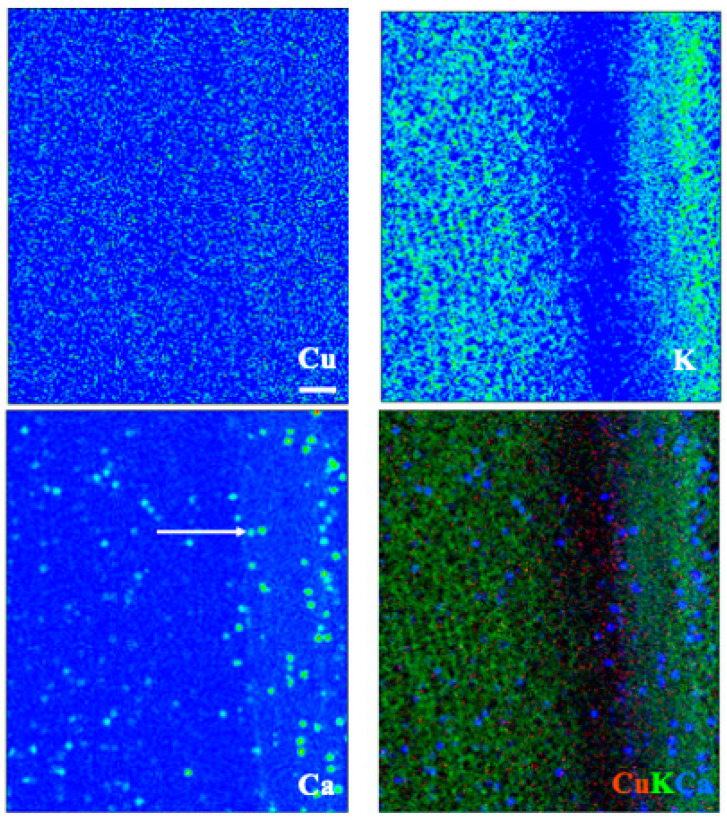
X-ray fluorescence images of a leaf of quillay at 100 μm Cu. Scale bar of 100 μm; arrow in lower left panel points to calcium oxalate crystals at the mid-rib.

**Figure 7 toxics-10-00237-f007:**
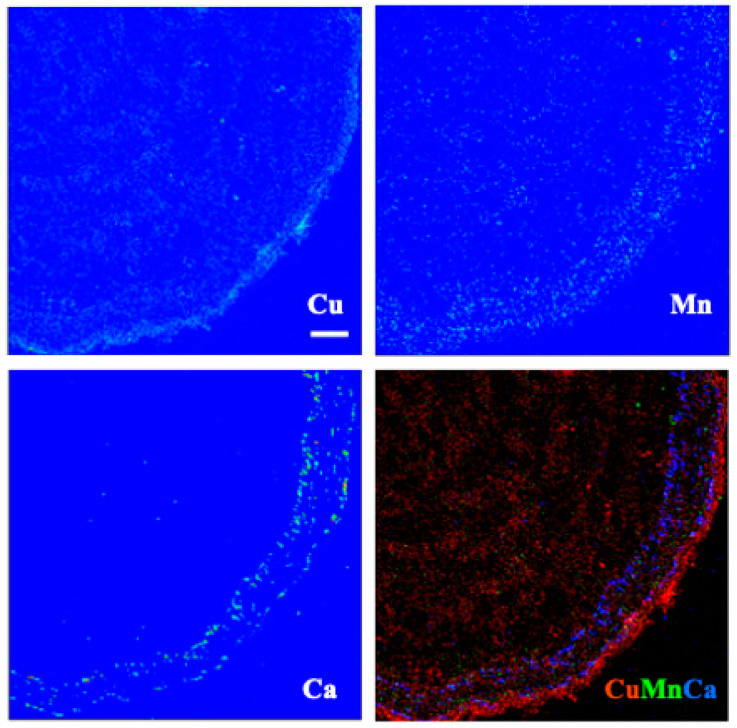
X-ray fluorescence images of an espino root at 100 μm Cu. Scale bar of 100 μm.

**Figure 8 toxics-10-00237-f008:**
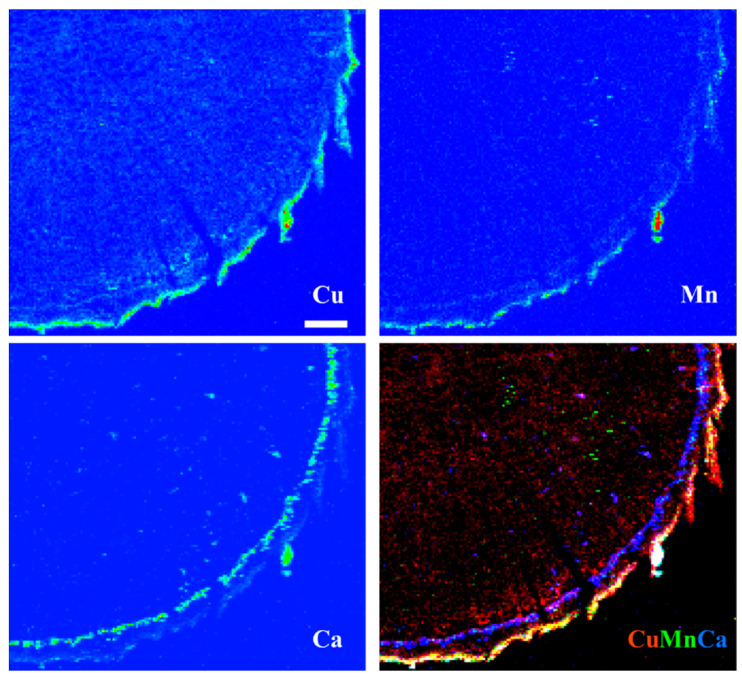
X-ray fluorescence images of a quillay root at 100 μm Cu. Scale bar of 100 μm.

**Figure 9 toxics-10-00237-f009:**
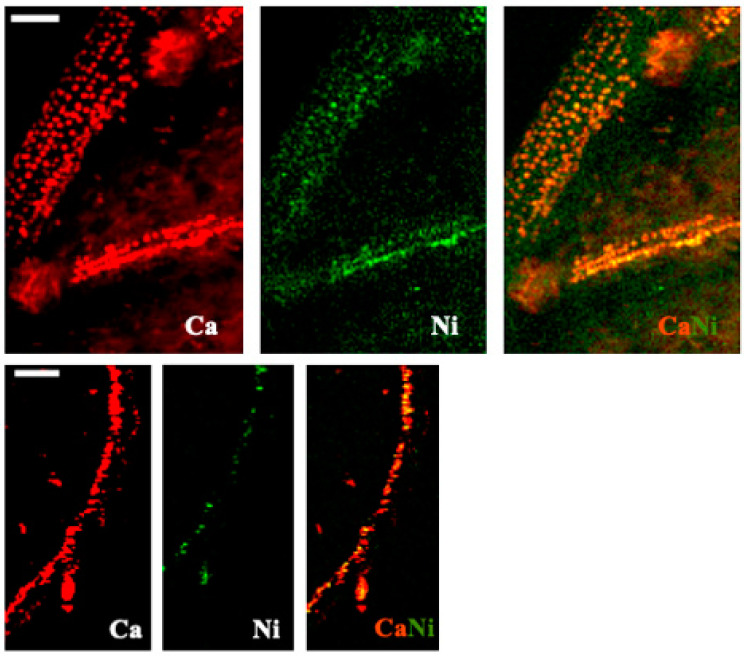
X-ray fluorescence images of Ca and Ni, and their combined distributions, in leaflets of espino (first row) and roots of quillay root (second row), both at 100 μm Cu. Scale bar of 100 μm.

**Table 1 toxics-10-00237-t001:** Descriptive statistics for heavy metals and elemental content in leaves and roots of espino and quillay in control, 50 and 100 μM Cu treatments.

Tissue	Species	Treatment	Ca (%)	Cu (mg/kg)	Fe (mg/kg)	K (%)	Mn (mg/kg)	Mo (mg/kg)	N (%)	Na (mg/kg)	P (%)	Zn (mg/kg)
			Mean	Mean	Mean	Mean	Mean	Mean	Mean	Mean	Mean	Mean
			[SD]	[SD]	[SD]	[SD]	[SD]	[SD]	[SD]	[SD]	[SD]	[SD]
Leaves	Espino	Control	19,000	3.3	49	20,000	44	1.6	3.7	25	2900	15
			--	--	--	--	--	--	--	--	--	--
		50 µM Cu	14,000	8.7	53	11,000	34	2.9	3.5	29	1400	18
			--	--	--	--	--	--	--	--	--	--
		100 µM Cu	14,000	13	51	12,000	55	1.0	3.2	20	1500	16
			--	--	--	--	--	--	--	--	--	--
	Quillay	Control	10,200	2.2	25.8	46,600	30	0.54	1.68	570	1860	15.8
			[2470]	[1.2]	[1.8]	[9840]	[13]	[0.2]	[0.1]	[697]	[152]	[2.2]
		50 µM Cu	12,800	10.1	25	43,200	48.6	1.2	1.6	320	2040	17.6
			[1480]	[3.7]	[6.3]	[2590]	[20.5]	[0.9]	[0.1]	[143]	[288]	[2.0]
		100 µM Cu	9740	9.4	20.8	36,600	52	0.9	1.6	119	1940	15
			[3680]	[4.8]	[4.9]	[6190]	[30.9]	[0.5]	[0.2]	[40.4]	[611]	[2.5]
Roots	Espino	Control	49,000	13	640	19,000	75	21	2.2	880	24,000	67
			--	--	--	--	--	--	--	--	--	--
		50 µM Cu	17,000	93	420	18,000	24	15	2.7	550	4400	25
			--	--	--	--	--	--	--	--	--	--
		100 µM Cu	30,000	170	410	15,000	79	21	2.3	610	13,000	40
			--	--	--	--	--	--	--	--	--	--
	Quillay	Control	16,200	12.2	158	19,600	52.6	7	1.6	1780	7620	42.8
			[1790]	[2.9]	[40.9]	[1820]	[14]	[2.2]	[0.3]	[497]	[1160]	[7.4]
		50 µM Cu	11,400	67.2	172	17,800	30.4	4.6	1.62	1840	4160	33
			[2040]	[14.5]	[59]	[3900]	[7]	[1]	[0.3]	[385]	[1080]	[7.4]
		100 µM Cu	14,600	96.6	126	20,800	63	4.72	1.54	1820	5980	37.6
			[1340]	[23.1]	[34.3]	[2680]	[24.3]	[1.7]	[0.3]	[536]	[1350]	[8.1]

N = 5 for quillay and espino (composite), SD: Standard Deviation, Ca: calcium, Cu: copper, Fe: iron, K: potassium, Mn: manganese, Mo: molybdenum, N: nitrogen, Na: sodium, P: phosphorus, Zn: zinc.

## Data Availability

Not applicable.

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
