# Peer review of "Enlightening the Pathway of Phytoremediation: Ecophysiology and X-ray Fluorescence Visualization of Two Chilean Hardwoods Exposed to Excess Copper"

_toxics, 2022, doi:10.3390/toxics10050237_

Round 1

Reviewer 1 Report

This is an interesting, well written paper describing well designed experiments.

Author Response

Response: Thank you very much for your support.

Reviewer 2 Report

The present study seems just presented the coarse results here but no comprehensive explanations, especially on the Cu tolerance pathways and implications. Some major issues should be improved as following:

1-Materials and methods-Plant material and treatment: Regarding to make Cu mix soil, how to mix Cu with soils to guaranttee homogenization on Cu, which is not clear. 

2-Materials and methods-Physiological measurements: Since it would be different on different leaves, how to choose the leaf for photochesmitry measurement?

3-How to define the studied plants as Cu tolerant species? Please provide the related standard.

4-If the research aim focused on phytoremediation, the total Cu accumulation via Cu levels multply biomass should be calculated and compare. Also, the Bioconcentration Factor and Tranfer Factot should be assessed. 

5-The present discussion is shallow. Please provide more discussion on the mechanissm for Cu distribution and accumulation. Moreover, why do these two plant species differ on Cu accumulation and distribution? What are the implications for the present study?

Author Response

Reviewer 2

The present study seems just presented the coarse results here but no comprehensive explanations, especially on the Cu tolerance pathways and implications. Some major issues should be improved as following:

1-Materials and methods-Plant material and treatment: Regarding to make Cu mix soil, how to mix Cu with soils to guaranttee homogenization on Cu, which is not clear. 

Response: It is doubtful that Cu concentrations were uniform throughout the soils in each pot, however, pots were always watered well-past the drip point and dripper outlets were re-positioned biweekly to promote a more even distribution (see additional text on lines 138-141). More importantly, we analyzed the soil mix from every pot at the termination of the experiment to establish the mean, median and range of Cu concentrations in the soil mix for each species in each treatment, as reported in Table S2. Overall, the Cu content of the soil mix averaged 15, 203 and 354 mg kg-1 in the 0 µM, 50 µM and 100 µM Cu treatments, respectively, as reported on Lines 204-205.

2-Materials and methods-Physiological measurements: Since it would be different on different leaves, how to choose the leaf for photochesmitry measurement?

Response: Thank you for noting this omission. We always selected the youngest mature leaf on each plant for gas exchange and subsequent fluorescence measurements. Since these measurements were conducted three times per plant over a five-week period, three such leaves per plant were sampled. The text has been amended to include this information (lines 148 and 156).

3-How to define the studied plants as Cu tolerant species? Please provide the related standard.

Response: As mentioned in the discussion (L363), the maximum allowable total Cu concentration in agricultural soils has been defined as 63 mg kg-1 and liquid media concentrations above 50 mM negatively impacts the growth of average plants (L354-357). Our 50 and 100 Cu-treatments greatly exceeded that amount and both of these plants were able to grow well on those enriched substrates. Moreover, espino actually benefited with the increased Cu treatment. We have added a reference to the effects of Cu on growth of beans as a further benchmark (line 355).

4-If the research aim focused on phytoremediation, the total Cu accumulation via Cu levels multply biomass should be calculated and compare. Also, the Bioconcentration Factor and Tranfer Factot should be assessed. 

Response: Phytoremediation can have different goals, which we now elaborate on lines 65-67. For example, plants that hyperaccumulate heavy metals in their aerial tissues are useful for phytoextraction and the removal of contaminants from sites. On the other hand, plants that tolerate heavy metal but keep them belowground are better for phytostabilization because they limit the spread of contaminants into wildlife or adjacent areas. Our paper explores Cu tolerance and uptake in two plants with excellent potential for phytostabilization. The calculation of how much Cu is contained in the total biomass (including the stems, which we did not analyze) is important from the perspective of phytoextraction and knowing how much Cu can be removed a site, whereas knowledge of Cu concentrations in tissues is more relevant in the context of phytostabilization and potential risks to higher trophic levels. Certainly, in both cases the Bcf and Tf are important and we now report them (as well as the bioaccumulation factor, Baf) as suggested by the reviewer (Methods: L 176-179, Results: L 267-275, Discussion: L 388-395).

5-The present discussion is shallow. Please provide more discussion on the mechanissm for Cu distribution and accumulation. Moreover, why do these two plant species differ on Cu accumulation and distribution? What are the implications for the present study?

Response:  It’s difficult to know why (as opposed to how) these two species differ in Cu accumulation and distribution. In general, however, different plant species have evolved different adaptations (strategies) for dealing with stress (i.e., there’s more than one way to solve any given problem). We have, however, added reference to strategies of avoidance and tolerance in the introduction (lines 67-77), and have added text to the discussion to interpret differences in Bcf, Bsf and Tf (lines 388-395). We also add reference to physiological mechanisms that may account for these differences (lines 457-469).

Reviewer 3 Report

Two typos. Line 236: in the caption of Figure 3 the references to figure b and c are exchanged. Line 439: Nickle instead of Nickel.

Author Response

Response: Thank you very much for your feedback, the figure and the word have been corrected.

Reviewer 4 Report

The manuscript presented describes how copper excess impacts two Chilean native hardwood species. Authors describes growth and photosynthesis parameters and elemental composition of leaves and roots of young plants after Cu exposure.  Both tree species used show potential for phytoremediation and the research presents considerable interest as a modern technology for sustainable development.

But why the introduction section and the abstract start with the global warming? Of course, it is a great challenge of the present and there are several relationships between the warming and Cu contamination of the environment. Authors should highlight those relationships or reformulate the beginning of the article and bring it into line with the aim and objectives of the study.

Also very interesting question, perhaps for further research: how plant species chosen will develop after several years of growing in soil with a high copper content.

Author Response

Reviewer 4

The manuscript presented describes how copper excess impacts two Chilean native hardwood species. Authors describes growth and photosynthesis parameters and elemental composition of leaves and roots of young plants after Cu exposure.  Both tree species used show potential for phytoremediation and the research presents considerable interest as a modern technology for sustainable development.

But why the introduction section and the abstract start with the global warming? Of course, it is a great challenge of the present and there are several relationships between the warming and Cu contamination of the environment. Authors should highlight those relationships or reformulate the beginning of the article and bring it into line with the aim and objectives of the study.

Response: Thanks for the note. We are stating that due to the energy transition, away from fossil fuels, global industrial demand for Cu will increase, therefore the tailings will also increase resulting in increased need for phytoremediation. We have added more detail on lines 16-17 to make that connection clearer.

Also very interesting question, perhaps for further research: how plant species chosen will develop after several years of growing in soil with a high copper content.

Response: We have previously studied these plants (and a few others) after 6 years of growth on unamended and amended tailings (Milla-Moreno & Guy. 2020. Growth response, uptake and mobilization of metals in native plant species on tailings at a Chilean copper mine. Int J Phytoremediation [doi: 10.1080/15226514.2020.1838435]). So, as noted in the introduction, the present manuscript actually follows up on the field study, by examining the effects of copper in the absence of other contaminating metals present in the tailings.  Results were more-or-less similar but there were some key differences between the two studies which we now highlight in the manuscript on lines 384-395.

Round 2

Reviewer 2 Report

Apparently, the manuscript was improved a lot, and most of my concerns were appropriately addressed. However, i am still lost in question 1 and some of question 4 and 5. 

Q1: The authors mentioned that the addition CuSO4 were added to achieve different levels of Cu treatment, but how to add and mix CuSO4 with soils, which is still not clear. 

Q4/5: Based on the calculated Bcf, Baf and Tf, the present two plants were Cu tolerant species, but not Cu accumulator because of <1 with Bcf, Baf and Tf. So i did not see their potential for Cu phytostablization or phytoextraction. then what is the implication for the present study? Please provide some crtitical thinking on it. 

Author Response

Reviewer 2

Apparently, the manuscript was improved a lot, and most of my concerns were appropriately addressed. However, I am still lost in question 1 and some of questions 4 and 5. 

Q1: The authors mentioned that the addition CuSO4 were added to achieve different levels of Cu treatment, but how to add and mix CuSO4 with soils, which is still not clear. 

Response:  As stated in the methods, we began exposing the plants to excess copper after they were established by adding CuSO4 to the fertigation solution. We did not mix it with the soils. Hence, all water that the plants received during the last 4 months of the experiment was contaminated with excess Cu at concentrations of 50 or 100 uM. We adjusted the text slightly on lines 138-140 to make this more clear. We also made corrections to lines 135-136 and Table S1.

Q4/5: Based on the calculated Bcf, Baf and Tf, the present two plants were Cu tolerant species, but not Cu accumulator because of <1 with Bcf, Baf and Tf. So i did not see their potential for Cu phytostablization or phytoextraction. then what is the implication for the present study? Please provide some crtitical thinking on it. 

Response: The factors that we previously established under field conditions had already indicated the potential of these two species for phytostabilization (i.e., Bcf  >1, Baf and Tf <0). The purpose of the present study was not to establish the Bcf and Baf in the greenhouse but rather to investigate the comparative physiology of espino and quillay and their internal responses to the provision of excess Cu (e.g., growth, photosynthesis and elemental distributions, including the Tf, which was similar to what we found in quillay in the field, but not in espino). The inclusion of the Tf calculation was a very useful addition that we thank the reviewer for. We don’t, however, consider the calculation of Bcf and Baf relative to the final concentrations of Cu in the soil mix to be relevant from an applied perspective, or even from a physiological perspective, as it is clear that much of the Cu was adsorbed by the organic matter when the fertigation solution passed through the pots. Calculation of the Bcf and Baf relative to the soil solution, which is what the plant roots actually experience, demonstrates the ability of both species to accumulate Cu in their root tissues in particular. We have added this information to the manuscript (lines 176-180, 270-278, 281-282) and adjusted the discussion accordingly (lines 377-379, 398-415).

Round 3

Reviewer 2 Report

Now my all concerns were celarly addressed. No more concern. Congratulations to all authors.